# $\text{A}^2$-NET: Learning Attribute-Aware Hash Codes for Large-Scale Fine-Grained Image Retrieval

**Xiu-Shen Wei**[1,2]**, Yang Shen**[1]**, Xuhao Sun**[1]**, Han-Jia Ye**[2]**, Jian Yang**[1]*

[1]Nanjing University of Science and Technology
[2]State Key Lab. for Novel Software Technology, Nanjing University

## Abstract

Our work focuses on tackling large-scale fine-grained image retrieval as ranking the images depicting the concept of interests (*i.e.*, the same sub-category labels) highest based on the fine-grained details in the query. It is desirable to alleviate the challenges of both fine-grained nature of small inter-class variations with large intra-class variations and explosive growth of fine-grained data for such a practical task. In this paper, we propose an Attribute-Aware hashing Network ($\text{A}^2$-NET) for generating attribute-aware hash codes to not only make the retrieval process efficient, but also establish explicit correspondences between hash codes and visual attributes. Specifically, based on the captured visual representations by attention, we develop an encoder-decoder structure network of a reconstruction task to unsupervisedly distill high-level attribute-specific vectors from the appearance-specific visual representations without attribute annotations. $\text{A}^2$-NET is also equipped with a feature decorrelation constraint upon these attribute vectors to enhance their representation abilities. Finally, the required hash codes are generated by the attribute vectors driven by preserving original similarities. Qualitative experiments on five benchmark fine-grained datasets show our superiority over competing methods. More importantly, quantitative results demonstrate the obtained hash codes can strongly correspond to certain kinds of crucial properties of fine-grained objects.

## 1 Introduction

Fine-grained image retrieval in computer vision aims to retrieve images belonging to multiple subordinate categories of a super-category (*aka* a meta-category), *e.g.*, different species of animals/plants [36], different models of cars [20], different kinds of retail products [39], etc. Its key challenge therefore lies with understanding fine-grained visual differences that sufficiently discriminate between objects that are highly similar in overall appearance, but differ in *fine-grained* features. Also, fine-grained retrieval still demands ranking all the instances so that images depicting the concept of interest (*e.g.*, the same sub-category label) are ranked highest based on the fine-grained details in the query.

In particular, with the explosive growth of fine-grained data in real applications [1, 14, 26, 36, 39], fine-grained hashing, as a promising solution for dealing with large-scale fine-grained retrieval tasks, has proven to be able to greatly reduce the storage cost and increase the query speed [8, 18] benefiting from the learned compact binary hash code representations. However, although previous works,

---

*Corresponding author. X.-S. Wei, Y. Shen, X. Sun and J. Yang are with PCA Lab, Key Lab of Intelligent Perception and Systems for High-Dimensional Information of Ministry of Education, and Jiangsu Key Lab of Image and Video Understanding for Social Security, School of Computer Science and Engineering, Nanjing University of Science and Technology. This work was supported by Natural Science Foundation of Jiangsu Province of China under Grant (BK20210340), the Fundamental Research Funds for the Central Universities (No. 30920041111), and CAAI-Huawei MindSpore Open Fund (CAAIXSJLJJ-2020-022A). The first two authors contributed equally to this work.

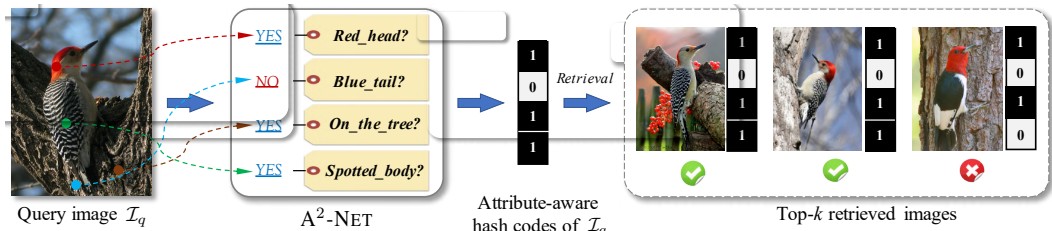

Query image $\mathcal{I}_q$      A²-NET      Attribute-aware hash codes of $\mathcal{I}_q$      Top-$k$ retrieved images

Figure 1: Key idea of our A²-NET, as well as the main process of fine-grained hashing based on our *attribute-aware* hash codes. In concretely, regarding a query image $\mathcal{I}_q$ of `Red bellied Woodpecker`, after returning all the correct results, a fine-grained image belonging to `Red headed Woodpecker` closest to the query image in terms of Hamming distance is also retrieved.

*e.g.*, [8, 18], achieved good retrieval performance, the bits of their hash codes correspond to no semantics, *i.e.*, *fine-grained attributes*. While, such attributes, *e.g.*, head color, tail color, male, female, living habits, are great means of describing fine-grained objects, in a way both humans and computers understand. In this paper, to establish an explicit correspondence between hash codes and visual attributes for not only further improving large-scale fine-grained retrieval accuracy, but more importantly integrating interpretation into deep learning based hash methods, we propose a unified Attribute-Aware hashing Network, termed as A²-NET (cf. Figure 1), for achieving these goals.

In our A²-NET, considering huge labor cost of supervised attribute annotations, we restrict ourselves in an unsupervised setting to automatically capture discriminative visual attributes from still images and then correspond the final learned hash code representations to these attributes. Therefore, a hash bit of learned hash codes could be both discriminative and intuitive. Additionally, thanks to the unsupervised setting, the attributes derived from A²-NET will be not restricted to pre-defined attributes like supervised-based attribute learning methods [15, 21, 42, 45]. Moreover, it can distill the most useful properties of fine-grained objects as attribute-aware hash codes in such an end-to-end trainable manner for accuracy retrieval among multiple similar subordinate categories.

More specifically, as the overall framework shown in Figure 2, our A²-NET consists of a fine-grained representation learning module and an attribute-aware hash codes generating module. It first leverages attention mechanisms to model fine-grained tailored patterns in terms of both global-level deep features $\boldsymbol{T}_i$ and local-level cues $\boldsymbol{T}_i^c$ from input image $\mathcal{I}_i$. Then, the appearance-specific features of these visual patterns $\boldsymbol{T}$ are aggregated and translated into semantic-specific representations $\boldsymbol{x}_i$. After that, we formulate the aforementioned unsupervised attribute learning as a reconstruction task of projecting $\boldsymbol{x}_i$ to an attribute vector $\boldsymbol{v}_i$ by performing an encoder-decoder structure network. Therefore, it can be expected that in the high-level attribute space, $\boldsymbol{v}_i$ could correspond to certain kinds of nameable properties of fine-grained objects. Moreover, a feature decorrelation constraint is further introduced upon $\boldsymbol{v}_i$ to both enhance the discriminative ability and remove the redundant correlation among these dimensions of attribute-specific features. Finally, our attribute-aware hash codes $\boldsymbol{u}_i$ are generated from $\boldsymbol{v}_i$ by conducting the hash code learning procedure.

To evaluate our model, we conduct extensive experiments using five benchmark fine-grained retrieval datasets for both accuracy and interpretability. Quantitative results of retrieval accuracy on these datasets show that the proposed A²-NET model obviously and consistently outperforms existing state-of-the-art methods. Qualitative visualization of the obtained attribute-aware hash codes demonstrates that these hash bits have strong correspondences to visual attributes of fine-grained objects (cf. Figure 4), even without employing attribute supervisions or part-level annotations. In addition, the ablation studies of these crucial components in A²-NET also validate their own effectiveness.

## 2 Related Work

### 2.1 Fine-Grained Image Retrieval

Fine-grained image retrieval as an integral part of fine-grained image analysis [41] has gained more and more traction in recent years [8, 25, 29, 40, 43, 47, 48]. What makes it challenging is that

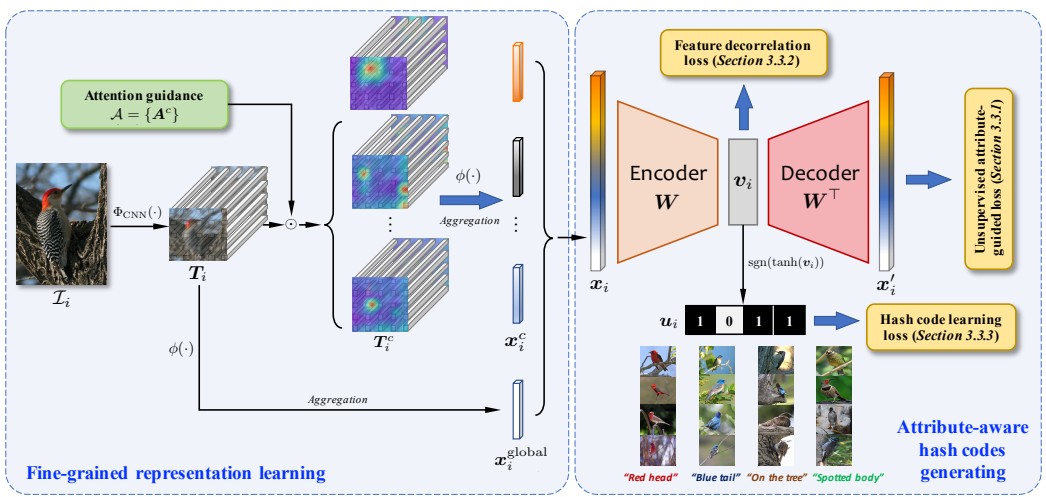

Figure 2: Overall framework of the proposed $A^2$-NET model, which consists of two crucial modules, *i.e.*, fine-grained representation learning and attribute-aware hash codes generating. The whole network can be end-to-end trainable, and is generally driven by the unsupervised attribute-guided reconstruction loss, the feature decorrelation loss and the hash code learning loss, cf. Section 3.3.

objects of fine-grained images have only subtle differences, and often largely vary in pose, scale, and orientation or can exhibit cross-modal differences (*e.g.*, sketch-based retrieval [25]).

Depending on the type of query image, the most studied areas of fine-grained image retrieval can be separated into two groups: fine-grained content-based image retrieval (FG-CBIR) and fine-grained sketch-based image retrieval (FG-SBIR). More specifically, in FG-CBIR, unsupervised learning based [40] and supervised learning based methods [44, 47, 48] were developed from different perspectives for handling fine-grained retrieval tasks, *e.g.*, localizing fine-grained parts [40], enhancing intra-class separability with inter-class compactness [48], and reducing the confidence of the fine-grained predictions [44], etc. While, FG-SBIR needs to not only capture fine-grained characteristics present in the sketches, but also possess the ability to traverse the sketch and image domain gap. In the literature of FG-SBIR, the earlier works, *e.g.*, [23, 43, 46], were mostly based on Siamese-triplet networks [3] to tackle the aforementioned challenges. Recently, some works tried to incorporate the advances of recent progress in self-supervised learning [6] and attention mechanisms [7] for further improving the retrieval accuracy of FG-SBIR, *e.g.*, [29, 32].

However, although these fine-grained retrieval methods achieved good results, they still have the limitations in the face of large-scale data, *i.e.*, the searching time for exact nearest neighbor is typically expensive or even impossible for the given queries. To alleviate this issue, fine-grained hashing, which aims to generate compact binary codes to represent fine-grained images, as a promising direction has attracted the attention in the fine-grained community very recently [8, 19]. More specifically, ExchNet [8] was the first to define the fine-grained hashing task and develop a fine-grained tailored method to firstly locate discriminative object parts and further learn binary hash codes for representing fine-grained images. In the same period, DSaH [19] was proposed to automatically mine salient regions and learn semantic-preserving hash codes simultaneously. Unfortunately, the learned hash bits of these methods lack any semantics which are more meaningful to fine-grained objects, and thus lack the model interpretability. Compared with them, our proposed $A^2$-NET can not only outperform the previous fine-grained hashing methods, but more importantly, the learned hash codes of $A^2$-NET are *attribute-aware*, *i.e.*, the hash bits of $A^2$-NET have strong correspondence to semantic visual properties that are useful for fine-grained image retrieval.

## 2.2 Learning to Hash

Hashing [38] is a widely-studied solution to approximate nearest neighbor search, which transforms the data item to a short code consisting of a sequence of bits (*i.e.*, hash codes). The research efforts of hashing can be categorized into two groups, including data-independent hashing (*aka* locality sensitive hashing [9, 27, 34]) and data-dependent hashing (*aka* learning to hash [4, 12, 17, 33]).

Specifically, locality sensitive hashing methods attempted to adjust hash learning from different perspectives, *e.g.*, the theory or machine learning views, to name a few: proposing random hash functions satisfying local sensitive property [9], developing better search schemes [27], providing faster computation of hash functions [34], etc. While, compared with locality sensitive hashing methods, since data-dependent hashing methods learn hash functions from a specific dataset to achieve similarity preserving, they can generally obtain superior retrieval accuracy. Especially for capitalizing on advances in deep learning, many well-performing methods were proposed to integrate feature learning and hash code learning into an end-to-end framework based on deep networks, *e.g.*, [4, 12, 17]. In particular, very recently researchers in the vision community have begun to pay attention to the more challenging and practical hashing task, *i.e.*, fine-grained hashing [8, 19]. To the best of our knowledge, this is the first work to equip these learned hash codes with strong correspondence to visual attributes for dealing with large-scale fine-grained image retrieval.

### 2.3 Visual Attributes

Attributes are typically mid-level semantic properties of objects [10], such as colors (*e.g.*, "red", "blue"), texture (*e.g.*, "striped", "spotted"), or even life habits of animals (*e.g.*, "living on the tree", "living in the water"). Visual attributes have exhibited their impact for strengthening various vision tasks, including facial verification [21], fine-grained categorization [45], zero-shot transfer [42], scene understanding [30], and so on. Most of the previous attribute learning methods are supervised by costly human-generated annotations and also are dependent on pre-defined attribute labels, *e.g.*, [15, 21, 42, 45]. In consequence, for large-scale problems, these supervised methods might be not feasible due to the restriction caused by the cumbersomely obtained attribute annotations. Moreover, even for some tasks, their visual attributes are quite hard to define. In this paper, to alleviate the aforementioned issues, we propose an $A^2$-NET model to formulate an *unsupervised learning* structure to project the learned visual features into an attribute space where it finally generates attribute-aware binary hash codes. Compared with previous attribute learning methods, our $A^2$-NET is independent with pre-defined attribute labels, and could automatically learn discriminative attribute-aware hash codes in a unified end-to-end trainable fashion. Furthermore, our method can not only correspond hash bits to visual attributes tailored for fine-grained objects, which shows significant improvements of retrieval accuracy, but also offer an intuitive way of deep hashing interpretation.

## 3 Methodology

In this section, we introduce the overall framework and notations of the proposed $A^2$-NET model, as well as elaborating the key modules of $A^2$-NET and its corresponding optimization algorithm.

### 3.1 Overall Framework and Notations

As illustrated in Figure 2, our $A^2$-NET model consists of two crucial modules, *i.e.*, a fine-grained representation learning module and an attribute-aware hash codes generating module. Given an input image $\mathcal{I}_i$, based on its corresponding deep activation tensor $\boldsymbol{T}_i \in \mathbb{R}^{C \times H \times W}$ extracted by a backbone CNN, a set of attention guidance $\mathcal{A} = \{\boldsymbol{A}^c\}$ is learned for capturing fine-grained tailored local patterns $\boldsymbol{T}_i^c$ from $\boldsymbol{T}_i$. To distill semantic cues and further generate the final attribute-aware binary hash codes, we propose to transform these appearance-specific features $\boldsymbol{T}$ towards semantic-specific representations $\hat{\boldsymbol{T}}$ by performing a transform network $\phi(\cdot)$. After aggregating $\hat{\boldsymbol{T}}$, the obtained attentive local-level features $\boldsymbol{x}_i^c$ are associated with the global-level feature $\boldsymbol{x}_i^{\text{global}}$ to form as a holistic feature representation $\boldsymbol{x}_i$. In order to generate attribute-aware binary hash codes, we conduct a reconstructing paradigm to project $\boldsymbol{x}_i$ as $\boldsymbol{v}_i$ in an attribute space where its data point corresponds to an attribute vector w.r.t. a certain kind of nameable properties of fine-grained objects (*e.g.*, "red head" or "spotted body"). Furthermore, with the aid of feature decorrelation, $\boldsymbol{v}_i$ is expected to be more discriminative by removing redundant correlation information. Finally, hash code learning is performed upon $\boldsymbol{v}_i$ to obtain the final attribute-aware binary codes $\boldsymbol{u}_i$.

### 3.2 Fine-Grained Representation Learning

Attention plays an important role in human perception [7, 16], and humans exploit a sequence of partial glimpses and selectively focus on salient parts of an object or a scene in order to better capture

visual structure [22]. Inspired by this, we incorporate the attention mechanism into representation learning to capture fine-grained local patterns for distinguishing subtle differences between these subordinate categories.

In concretely, we extract the deep feature of its input image $\mathcal{I}_i$ via a backbone CNN model $\Phi_{\mathrm{CNN}}(\cdot)$ by

$$\boldsymbol{T}_i = \Phi_{\mathrm{CNN}}(\mathcal{I}_i) \in \mathbb{R}^{C \times H \times W} . \tag{1}$$

Then, based on $\boldsymbol{T}_i$, $C$ attention guidance $\boldsymbol{A}^c \in \mathbb{R}^{H \times W}$ is generated as a set of attention maps, *i.e.*, $\mathcal{A}$. The attention guidance $\boldsymbol{A}^c$ is designed to evaluate which deep descriptors [40] in these $H \times W$ cells should be attended or even overlooked by conducting

$$\boldsymbol{T}_i^c = \boldsymbol{A}^c \odot \boldsymbol{T}_i , \tag{2}$$

where $\odot$ is the element-wise Hadamard product. To obtain the final attribute-aware binary codes, it is desirable to transform these appearance-specific (*i.e.*, low-level) features $\boldsymbol{T}$ to semantic-specific (*i.e.*, mid-level) representations which are closer to the attribute space. Thus, a transforming network $\phi(\cdot)$, which is equipped with a stack of convolution layers, is performed on $\boldsymbol{T}$ as follows:

$$\hat{\boldsymbol{T}}_i^c = \phi(\boldsymbol{T}_i^c; \theta_{\mathrm{local}}) , \tag{3}$$
$$\hat{\boldsymbol{T}}_i = \phi(\boldsymbol{T}_i; \theta_{\mathrm{global}}) , \tag{4}$$

where $\theta$ presents the parameters of the corresponding transforming networks w.r.t. $\boldsymbol{T}_i^c$ and $\boldsymbol{T}_i$, respectively. Then, we aggregate $\hat{\boldsymbol{T}}_i^c$ and $\hat{\boldsymbol{T}}_i$ by conducting global average-pooling and correspondingly obtain the attentive local-level features $\boldsymbol{x}_i^c$, as well as the global-level feature $\boldsymbol{x}_i^{\mathrm{global}}$. The holistic feature representation w.r.t. the input image $\mathcal{I}_i$ is achieved by concatenating both $\boldsymbol{x}_i^c$ and $\boldsymbol{x}_i^{\mathrm{global}}$, *i.e.*, $\boldsymbol{x}_i = \left[ \boldsymbol{x}_i^c; \boldsymbol{x}_i^{\mathrm{global}} \right] = F(\mathcal{I}_i; \Theta) \in \mathbb{R}^d$. Note that, we hereby abstract the aforementioned fine-grained feature learning process as a function $F(\mathcal{I}_i; \Theta)$ associated with its parameters $\Theta$.

### 3.3 Attribute-Aware Hash Codes Generating

How to generate attribute-aware hash codes is the key of our $\mathrm{A}^2$-NET model. We elaborate it in the following three aspects, *i.e.*, unsupervised attribute-guided learning, attribute-specific feature decorrelation, and hash code learning.

#### 3.3.1 Unsupervised Attribute-Guided Learning

In real-applications, especially for the large-scale and fine-grained tasks, attribute annotations are always infeasible, which limits the learning process to be conducted in an unsupervised setting. While, in the literature, the main goal of unsupervised learning is to capture regularities in data for the purpose of extracting useful representations or for restoring corrupted data [31]. Many unsupervised methods explicitly produce *internal latent units or codes*, from which the data is to be reconstructed.

Inspired by this, we develop an unsupervised attribute-guided reconstruction component to project the holistic representation $\boldsymbol{x}_i$ of $\mathcal{I}_i$ into a latent space, *i.e.*, the attribute space $\mathcal{V}$. In $\mathcal{V}$, its high-level vectors are designed to have certain desirable properties, *e.g.*, corresponding to semantic properties of fine-grained objects (*aka* "fine-grained attributes").

More specifically, in our $\mathrm{A}^2$-NET, the unsupervised attribute-guided learning is realized by a reconstruction paradigm with an encoder-decoder structure, as shown in Figure 2. In concretely, given a batch of $n$ training data $\mathcal{I}_i$, their holistic representations $\boldsymbol{X} = \{\boldsymbol{x}_1; \boldsymbol{x}_2; \ldots; \boldsymbol{x}_n\} \in \mathbb{R}^{d \times n}$ can be obtained as aforementioned. By formulation, the encoder projects $\boldsymbol{X}$ into the attribute space $\mathcal{V}$ with a projection matrix $\boldsymbol{W} \in \mathbb{R}^{k \times d}$ to get an internal latent representation $\boldsymbol{V} \in \mathbb{R}^{k \times n}$ w.r.t. $\boldsymbol{X}$. In particularly, we set that the dimension of latent representation $k$ equals the number of hash bits in the final binary hash code $\boldsymbol{u}_i$. Furthermore, each column of $\boldsymbol{V}$, *i.e.*, $\boldsymbol{v}_i \in \mathbb{R}^k$, can derive $\boldsymbol{u}_i$ by

$$\boldsymbol{u}_i = \mathrm{sgn}(\tanh(\boldsymbol{v}_i)) . \tag{5}$$

Meanwhile, regarding $\boldsymbol{v}_i$, we also consider to reconstruct its input $\boldsymbol{x}_i$ by a decoder as a counterpart of the encoder. Therefore, on one hand, such a reconstruction paradigm can reduce and further distill high-level semantic cues in the attribute space $\mathcal{V}$. While, on the other hand, it can drive the training

of $A^2$-NET by preserving the similarity between queried hash codes and database points in terms of hash code learning (cf. Section 3.3.3).

In concretely, the learning objective of unsupervised attribute-guided reconstruction is written as follows:

$$\min_{\boldsymbol{W}} \|\boldsymbol{X} - \boldsymbol{W}^\top \boldsymbol{W} \boldsymbol{X}\|_F^2 \quad \text{s.t. } \boldsymbol{W} \boldsymbol{X} = \boldsymbol{V}' = \tanh(\boldsymbol{V}), \tag{6}$$

where the decoder (*i.e.*, a counterpart of the encoder) is realized by $\boldsymbol{W}^\top$ to simplify the network. However, directly minimizing Eq. (6) with a hard constraint is difficult to optimize. Therefore, we relax the constraint into a soft constraint, and then the learning objective can be rewritten as

$$\min_{\boldsymbol{W}} \|\boldsymbol{X} - \boldsymbol{W}^\top \boldsymbol{V}'\|_F^2 + \lambda \|\boldsymbol{W} \boldsymbol{X} - \boldsymbol{V}'\|_F^2. \tag{7}$$

### 3.3.2 Attribute-Specific Feature Decorrelation

By conducting the aforementioned unsupervised attribute-guided learning, we can obtain the internal latent vectors $\boldsymbol{V}'$ as the attribute-specific features. In order to both enhance the discriminative ability and remove the redundant correlation among these attribute-specific features, we introduce a feature decorrelation constraint upon $\boldsymbol{V}'$, which is formulated by

$$\min_{\boldsymbol{V}'} \|\boldsymbol{V}'\boldsymbol{V}'^\top - n\boldsymbol{I}\|_F^2, \tag{8}$$

where $\boldsymbol{I}$ is the identity matrix and $n$ is the batch size. Such a feature decorrelation constraint is preferable to construct independent features and reduce redundant information. Therefore, based on both unsupervised attribute-guided reconstruction and attribute-specific feature decorrelation, the final learned hash codes are expected to be both attribute-aware and hash-bit independent.

### 3.3.3 Hash Code Learning

In the following, we conduct the hash code learning based on the obtained attribute-specific features. Assume that we have $n$ query data points which are denoted as $\{\boldsymbol{q}_i\}_{i=1}^n$, as well as $m$ database points which are denoted as $\{\boldsymbol{v}_j\}_{j=1}^m$. Note that, both $\boldsymbol{q}_i$ and $\boldsymbol{v}_i$ are belonging to the attribute space $\mathcal{V}$. By following Eq. (5), the corresponding binary hash codes can be obtained via

$$\boldsymbol{u}_i = \text{sgn}(\tanh(\boldsymbol{q}_i)), \tag{9}$$
$$\boldsymbol{z}_j = \text{sgn}(\tanh(\boldsymbol{v}_j)), \tag{10}$$

where $\boldsymbol{u}_i, \boldsymbol{z}_j \in \{-1, +1\}^k$. The goal of our hash code learning is to learn binary hash codes for both query points and database points from $\{\boldsymbol{q}_i\}_{i=1}^n$, $\{\boldsymbol{v}_j\}_{j=1}^m$, and the pairwise supervised information, *i.e.*, $\boldsymbol{S} \in \{-1, +1\}^{n \times m}$. To preserve the pairwise similarity, we adopt the $\ell_2$ loss between the supervised information (*aka* similarity) and the inner product of query-database point binary code pairs. It can be formulated as follows:

$$\min_{\boldsymbol{U}, \boldsymbol{Z}} \sum_{i=1}^n \sum_{j=1}^m \left(\boldsymbol{u}_i^\top \boldsymbol{z}_j - k S_{ij}\right)^2$$
$$\text{s.t.} \quad \boldsymbol{U} \in \{-1, +1\}^{n \times k}, \boldsymbol{Z} \in \{-1, +1\}^{m \times k}. \tag{11}$$

Overall, we get the final objective of the proposed $A^2$-NET model by considering Eq. (7), Eq. (8) and Eq. (11) together as follows:

$$\min_{\boldsymbol{W}, \boldsymbol{\Theta}} \mathcal{L}(\mathcal{I}) = \|\boldsymbol{X} - \boldsymbol{W}^\top \boldsymbol{V}'\|_F^2 + \lambda \|\boldsymbol{W}\boldsymbol{X} - \boldsymbol{V}'\|_F^2 + \alpha \|\boldsymbol{V}'\boldsymbol{V}'^\top - n\boldsymbol{I}\|_F^2 + \beta \sum_{i=1}^n \sum_{j=1}^m \left(\boldsymbol{u}_i^\top \boldsymbol{z}_j - k S_{ij}\right)^2, \tag{12}$$

where $\lambda$, $\alpha$ and $\beta$ are hyper-parameters as the trade-off.

In practice, during training, it might be only available a set of database points $\{\boldsymbol{v}_j\}_{j=1}^m$ without query points. Thus, we randomly sample $n$ data points from database to construct a query set, and denote the indices of all the database points as $\Gamma$ with the indices of the query set as $\Omega$. Additionally, because we cannot back-propagate the gradient to $\Theta$ due to the $\text{sgn}(\cdot)$ function, we omit the $\text{sgn}(\cdot)$ function and only apply $\tanh(\cdot)$ for relaxation in Eq. (10) of the whole optimization process. Therefore, the

optimization formulation of $A^2$-NET can be rewritten with only database points $\{\boldsymbol{v}_j\}_{j=1}^m$ for training as:

$$\min_{\boldsymbol{W},\boldsymbol{\Theta}} \mathcal{L}(\mathcal{I}) = \|\boldsymbol{X} - \boldsymbol{W}^\top \boldsymbol{V}'\|_F^2 + \lambda\|\boldsymbol{W}\boldsymbol{X} - \boldsymbol{V}'\|_F^2 + \alpha\|\boldsymbol{V}'\boldsymbol{V}'^\top - n\boldsymbol{I}\|_F^2$$
$$+ \beta \sum_{i\in\Omega}\sum_{j\in\Gamma} \left(\tanh(\boldsymbol{W} \cdot F(\mathcal{I}_i;\Theta))^\top \boldsymbol{z}_j - kS_{ij}\right)^2 . \qquad (13)$$

For optimization, our $A^2$-NET does not require complicated two-stage learning algorithms, *e.g.*, the alternative optimization strategy. In experiments, we employ the back-propagation algorithm and follow [17] to train the whole $A^2$-NET model in a unified end-to-end manner.

### 3.4 Out-of-Sample Extension

After training $A^2$-NET, the learned model can be applied for generating binary codes for query points including unseen query points in the training phase. Specifically, we can use the following equation to generate the binary code for $\mathcal{I}_q$:

$$\boldsymbol{u}_q = \text{sgn}(\tanh(\boldsymbol{W} \cdot F(\mathcal{I}_q;\Theta))) . \qquad (14)$$

## 4 Experiments

### 4.1 Datasets

By following ExchNet [8], our experiments are conducted on five fine-grained benchmark datasets, *i.e.*, *CUB200-2011* [37], *Aircraft* [28], *Food101* [2], *NABirds* [35] and *VegFru* [14]. Specifically, *CUB200-2011* is one of the most popular used fine-grained datasets. It contains 11,788 bird images from 200 bird species and is officially split into 5,994 images for training and 5,794 images for test. *Aircraft* contains 10,000 images spanning 100 aircraft models with 3,334 for training, 3,333 for validation and 3,333 for test. For large-scale datasets, *Food101* contains 101 kinds of foods with 101,000 images, where for each class, 250 test images are checked manually for correctness while 750 training images still contain a certain amount of noises. *NABirds* is a high quality dataset which has 48,562 images of North American birds with 555 sub-categories, where 23,929 for training with 24,633 for test. *VegFru* is another large-scale fine-grained dataset covering 200 kinds of vegetables and 92 kinds of fruits with 29,200 for training, 14,600 for validation and 116,931 for test.

### 4.2 Baselines and Implementation Details

**Baselines** In experiments, we compare our proposed model to the following competitive baselines, *i.e.*, ITQ [11], SDH [33], DPSH [24], HashNet [5], and ADSH [17]. Among them, DPSH, HashNet and ADSH are deep learning based methods, while ITQ and SDH are not. Furthermore, we also compare the results of our $A^2$-NET with state-of-the-arts of fine-grained hashing methods, including ExchNet [8]. Additionally, another fine-grained hashing method, *i.e.*, DSaH [19], also achieved good retrieval accuracy. However, due to its empirical settings quite distant from other existing fine-grained hashing methods, for fair comparisons, we strictly control empirical settings as the same as those of [19] and compare the results of our $A^2$-NET with its results in the supplementary materials.

**Implementation Details** For fair comparisons, we follow the efficient training setting in Exch-Net [8]. In concretely, for *CUB200-2011*, *Aircraft* and *Food101*, we sample 2,000 images per epoch, while 4,000 samples are randomly selected for *NABirds* and *VegFru*. For the training details, regarding the backbone model, we can choose any network structure as the base network for the fine-grained representation learning module. While, by following [8], ResNet-50 [13] is employed in experiments. The total number of training epochs is 20, and the number of batch size is set as 16. While, different from ExchNet, our model only requires a smaller iteration time until convergence. Specifically, for these datasets containing less than 20,000 training images, the iteration time $T_{\max}$ is 60, and the learning rate is divided by 10 at the $50^{\text{th}}$ iteration. For other datasets, $T_{\max}$ is set as 70, and the learning rate is divided by 10 at the $60^{\text{th}}$ iteration. The hyper-parameters, *i.e.*, $\lambda$, $\alpha$ and $\beta$ in Eq. (13), are set as 1, $\frac{1}{n\times k}$ and $\frac{12}{k}$, respectively. The number of attention guidance equals the number of hash bits. The optimizer is standard mini-batch stochastic gradient descent with the weight decay as $10^{-4}$. All experiments are conducted with a GeForce RTX 2080 Ti GPU.

Table 1: Comparisons of retrieval accuracy (% mAP) on five fine-grained benchmark datasets.

| Datasets | # bits | ITQ | SDH | DPSH | HashNet | ADSH | ExchNet | **Ours** |
|---|---|---|---|---|---|---|---|---|
| *CUB200-2011* | 12 | 6.80 | 10.52 | 8.68 | 12.03 | 20.03 | 25.14 | **33.83** |
| | 24 | 9.42 | 16.95 | 12.51 | 17.77 | 50.33 | 58.98 | **61.01** |
| | 32 | 11.19 | 20.43 | 12.74 | 19.93 | 61.68 | 67.74 | **71.61** |
| | 48 | 12.45 | 22.23 | 15.58 | 22.13 | 65.43 | 71.05 | **77.33** |
| *Aircraft* | 12 | 4.38 | 4.89 | 8.74 | 14.91 | 15.54 | 33.27 | **40.00** |
| | 24 | 5.28 | 6.36 | 10.87 | 17.75 | 23.09 | 45.83 | **63.66** |
| | 32 | 5.82 | 6.90 | 13.54 | 19.42 | 30.37 | 51.83 | **72.51** |
| | 48 | 6.05 | 7.65 | 13.94 | 20.32 | 50.65 | 59.05 | **81.37** |
| *Food101* | 12 | 6.46 | 10.21 | 11.82 | 24.42 | 35.64 | 45.63 | **46.44** |
| | 24 | 8.20 | 11.44 | 13.05 | 34.48 | 40.93 | 55.48 | **66.87** |
| | 32 | 9.70 | 13.36 | 16.41 | 35.90 | 42.89 | 56.39 | **74.27** |
| | 48 | 10.07 | 15.55 | 20.06 | 39.65 | 48.81 | 64.19 | **82.13** |
| *NABirds* | 12 | 2.53 | 3.10 | 2.17 | 2.34 | 2.53 | 5.22 | **8.20** |
| | 24 | 4.22 | 6.72 | 4.08 | 3.29 | 8.23 | 15.69 | **19.15** |
| | 32 | 5.38 | 8.86 | 3.61 | 4.52 | 14.71 | 21.94 | **24.41** |
| | 48 | 6.10 | 10.38 | 3.20 | 4.97 | 25.34 | 34.81 | **35.64** |
| *VegFru* | 12 | 3.05 | 5.92 | 6.33 | 3.70 | 8.24 | 23.55 | **25.52** |
| | 24 | 5.51 | 11.55 | 9.05 | 6.24 | 24.90 | 35.93 | **44.73** |
| | 32 | 7.48 | 14.55 | 10.28 | 7.83 | 36.53 | 48.27 | **52.75** |
| | 48 | 8.74 | 16.45 | 9.11 | 10.29 | 55.15 | 69.30 | **69.77** |

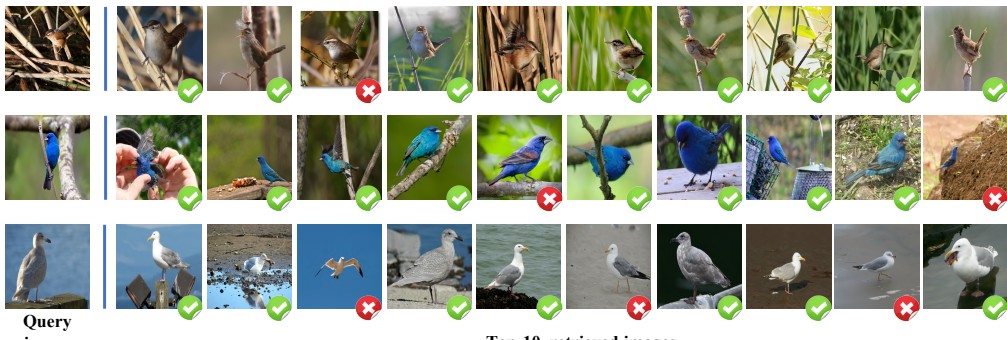

**Query image**    **Top-10 retrieved images**

Figure 3: Examples of top-10 retrieved images on *CUB200-2011* of 48-bit hash codes by our $A^2$-NET.

### 4.3 Main Results

Table 1 presents the mean average precision (mAP) results of fine-grained retrieval on these five aforementioned fine-grained benchmark datasets. For each dataset, we report the results of four lengths of hash bits, *i.e.*, 12, 24, 32, and 48, for evaluations. As shown in that table, our proposed $A^2$-NET model significantly and consistently outperforms the other baseline methods on these datasets. In particular, compared with the state-of-the-art method ExchNet [8], our $A^2$-NET achieves 17.83% and 17.88% improvements over ExchNet of 24-bit and 32-bit experiments on *Aircraft* and *Food-101*, respectively. Moreover, $A^2$-NET also obtains superior results with an absolute value of about 80% mAP on *CUB200-2011*, *Aircraft* and *Food101* with 48-bit hash codes. These observations validate the effectiveness of the proposed $A^2$-NET model, as well as its promising practicality in real-applications of fine-grained retrieval. Additionally, in Figure 3, we illustrate several retrieval results on *CUB200-2011*, which shows that $A^2$-NET can retrieve well among multiple subordinate categories when the same species of birds with diverse variations appear in different kinds of background. Also, there also exist several failure cases, where quite tiny differences (*e.g.*, caused by different views) between the query image and the returned images are demanded by carefully observations.

### 4.4 Ablation Studies

In this section, we demonstrate the effectiveness of these crucial components of the proposed $A^2$-NET model, *i.e.*, the attention-based fine-grained representation learning component 3.2, the unsupervised

Table 2: Retrieval accuracy (% mAP) with incremental components of the proposed A$^2$-NET model.

| Configurations | CUB200-2011 | | | | Food101 | | | |
|---|---|---|---|---|---|---|---|---|
| | 12 bits | 24 bits | 32 bits | 48 bits | 12 bits | 24 bits | 32 bits | 48 bits |
| Vanilla backbone | 20.03 | 50.33 | 61.68 | 65.43 | 35.64 | 40.93 | 42.89 | 48.81 |
| + Attention (Sec. 3.2) | 27.42 | 58.17 | 68.24 | 76.10 | 41.33 | 65.07 | 70.06 | 78.51 |
| + Reconstruction (Sec. 3.3.1) | 33.31 | 60.65 | 71.28 | 77.10 | 45.02 | **67.49** | 73.57 | 81.63 |
| + Feature decorrelation (Sec. 3.3.2) | **33.83** | **61.01** | **71.61** | **77.33** | **46.44** | 66.87 | **74.27** | **82.13** |

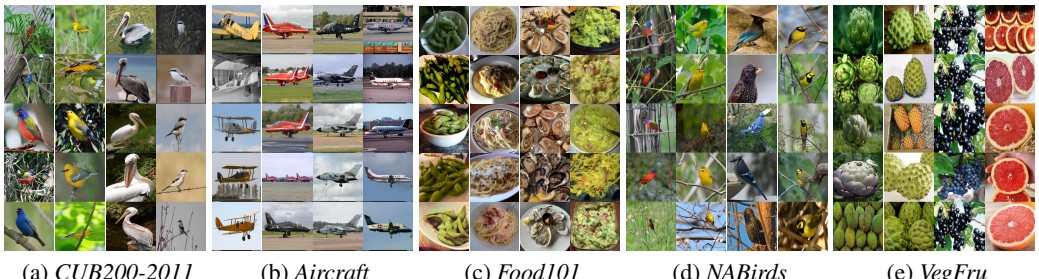

(a) *CUB200-2011*  (b) *Aircraft*  (c) *Food101*  (d) *NABirds*  (e) *VegFru*

Figure 4: Quality demonstrations of the learned attribute-aware hash codes by the proposed A$^2$-NET model. Each column in each sub-figure can strongly correspond to a certain kind of properties of the fine-grained objects, *e.g.*, "yellow birds in the forest", "double-winged aircrafts", "noodle-like food", "Bromeliaceae fruits", etc. (Best viewed in color and zoomed in.)

attribute-guided reconstruction component (cf. Section 3.3.1) and the attribute-specific feature decorrelation component (cf. Section 3.3.2). In the ablation studies, we apply these components incrementally on a vanilla backbone (*i.e.*, ResNet-50) as the baseline. As evaluated in Table 2, by stacking these two components one by one, the retrieval results are steadily improved, which justifies the effectiveness of our proposed components in A$^2$-NET.

### 4.5 Qualitative Analyses of Attribute-Aware Hash Codes

We hereby discuss the quality of the learned attribute-aware hash codes $u_i$ of A$^2$-NET. After obtaining $u_i$, we visualize fine-grained images retrieved by a random single hash bit of $u_i$ to demonstrate the strong correspondence between visual attributes and the obtained hash bits. All the five datasets in experiments are used as examples to illustrate the quality. As observed in Figure 4, images of each column have some similar fine-grained object properties, *i.e.*, visual attributes. Indeed, the learned hash codes are apparently attribute-aware, which could provide an explanation of the A$^2$-NET's success in fine-grained retrieval. Meanwhile, it also offers human-understandable interpretation for such a deep learning based fine-grained hashing method.

## 5 Conclusion

In this paper, we proposed an Attribute-Aware hashing Network, *i.e.*, A$^2$-NET, for dealing with the large-scale fine-grained image retrieval task. Particularly, A$^2$-NET was designed as expected to be efficient, effective and more importantly interpretable. In concretely, by developing an unsupervised attribute-guided reconstruction method based on the obtained appearance-specific visual representation with attention, it can distill attribute-specific vectors in a high-level attribute space. After further performing feature decorrelation upon attribute-specific vectors, their discriminative ability is strengthened for representing a fine-grained object. Then, hash codes can be generated from these attribute-specific vectors and thus became attribute-aware. Both qualitative and quantitative experiments demonstrate the effectiveness of our A$^2$-NET. Additionally, visual attributes have been shown to be useful in describing both known and unknown entities, which motivates us to study identifying unobserved sub-categories, *i.e.*, zero-shot fine-grained recognition, based on our attribute-aware hash codes as the future work.

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
