# A$^2$-NET: Learning Attribute-Aware Hash Codes for Large-Scale Fine-Grained Image Retrieval (Supplementary Materials)

**Xiu-Shen Wei**[1,2]**, Yang Shen**[1]**, Xuhao Sun**[1]**, Han-Jia Ye**[2]**, Jian Yang**[1]

[1]Nanjing University of Science and Technology
[2]State Key Lab. for Novel Software Technology, Nanjing University

In the supplementary materials, we present further information about the proposed A$^2$-NET model, including: 1) Additional experimental results of other comparison methods, especially DSaH [4]; 2) More examples of retrieved results on other fine-grained benchmark datasets.

## 1 Additional experimental results of other comparison methods

Apart from ExchNet, DSaH [4] is another fine-grained hashing method which has achieved good retrieval accuracy. For fair comparisons, we strictly control empirical settings as the same as those of [4] and compare the results of our A$^2$-NET with its results and three following methods, *i.e.*, DPSH [6], DTQ [7] and HBMP [2].

Specifically, we follow the settings of DSaH [4] and conduct experiments on two fine-grained datasets, *i.e.*, *Stanford Dogs* [5] and *CUB200-2011* [10]. In concretely, *Stanford Dogs* consists of 20,580 images in 120 classes while each class contains about 150 images. The dataset is divided into the train set (100 images per class) and the test set (totally 8,580 images for all categories). *CUB200-2011* contains 11,788 bird images from 200 bird species and is officially split into 5,994 images for training and 5,794 images for test. We use AlexNet as backbone and it is not fine-tuned on each dataset.

As shown in Table 1, our A$^2$-NET significantly outperforms the other baseline methods on these two datasets by following the same settings of [4]. In particular, compared with DSaH [4], our A$^2$-NET achieves 10% and 7% improvements on *Stanford Dogs* and *CUB200-2011* in average.

Table 1: Comparisons of retrieval accuracy (% mAP) on two benchmark fine-grained datasets.

| Methods | *Stanford Dogs* | | | | *CUB200-2011* | | | |
|---|---|---|---|---|---|---|---|---|
| | 12 bits | 24 bits | 36 bits | 48 bits | 12 bits | 24 bits | 36 bits | 48 bits |
| DPSH [6] | 17.7 | 22.1 | 26.5 | 31.5 | 7.2 | 7.6 | 8.4 | 7.9 |
| DTQ [7] | 18.5 | 18.7 | 18.7 | 18.8 | 7.3 | 11.3 | 15.4 | 18.3 |
| HBMP [2] | 19.0 | 23.8 | 28.7 | 32.8 | 8.9 | 10.9 | 14.2 | 16.8 |
| DSaH [4] | 24.4 | 28.7 | 36.3 | 40.8 | 14.2 | 20.9 | 23.2 | 28.5 |
| **Ours** | **36.6** | **44.8** | **46.9** | **47.4** | **19.2** | **27.2** | **32.5** | **36.7** |

## 2 More examples of retrieved results on other fine-grained datasets

We present more retrieval results on *Aircraft* [8], *Food101* [1], *NABirds* [9] and *VegFru* [3]. As shown in the following figures, our proposed A$^2$-NET can retrieve well among multiple subordinate categories. There also exist several failure cases, where quite tiny differences (*e.g.*, caused by different views) between the query image and the returned images are demanded by carefully observations.

35th Conference on Neural Information Processing Systems (NeurIPS 2021).

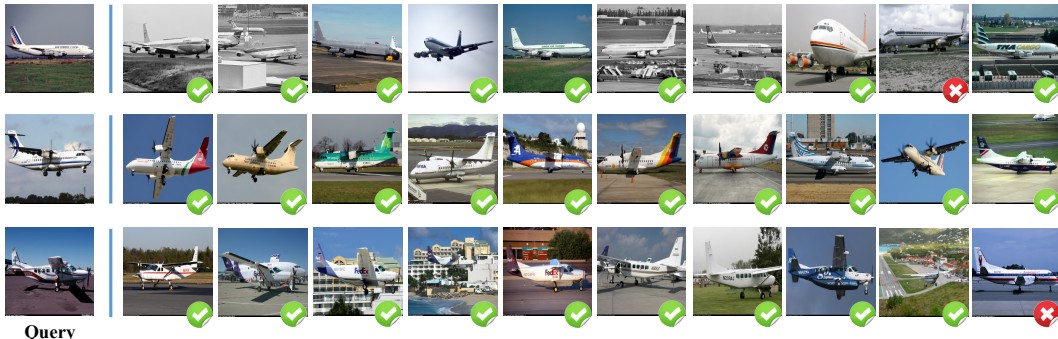

**Query image**

**Top-10 retrieved images**

Figure 1: Examples of top-10 retrieved images on *Aircraft* of 48-bit hash codes by our A$^2$-NET.

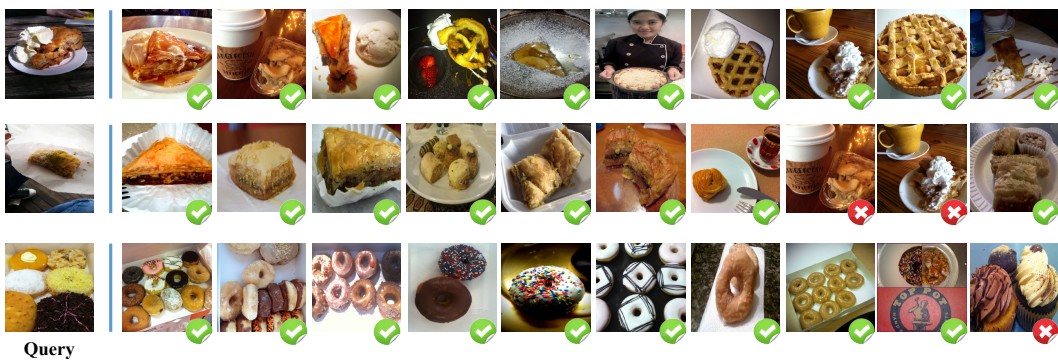

**Query image**

**Top-10 retrieved images**

Figure 2: Examples of top-10 retrieved images on *Food101* of 48-bit hash codes by our A$^2$-NET.

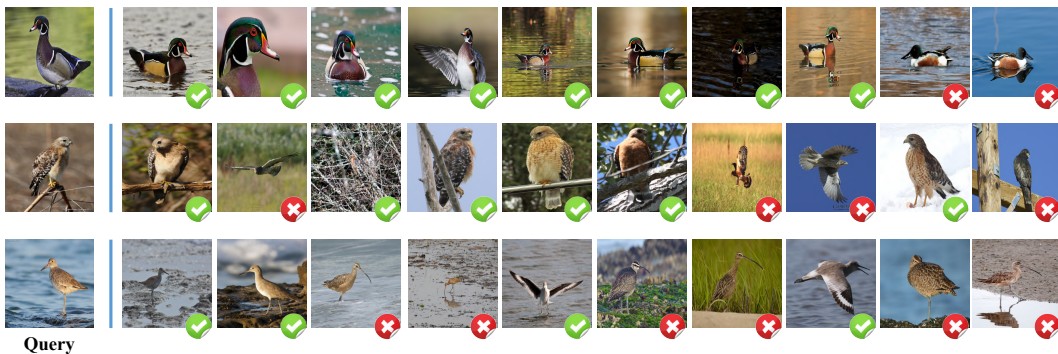

**Query image**

**Top-10 retrieved images**

Figure 3: Examples of top-10 retrieved images on *NABirds* of 48-bit hash codes by our A$^2$-NET.

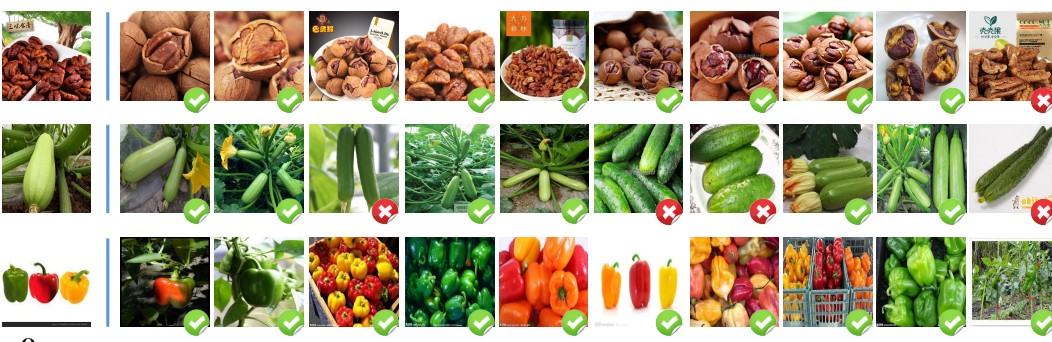

**Query image**

**Top-10 retrieved images**

Figure 4: Examples of top-10 retrieved images on *VegFru* of 48-bit hash codes by our A$^2$-NET.