# OpenReview forum: "A$^2$-Net: Learning Attribute-Aware Hash Codes for Large-Scale Fine-Grained Image Retrieval"
_NeurIPS.cc/2021/Conference — NeurIPS 2021 Spotlight_

### Official Review · Reviewer_gTRX · 2021-07-14

**Rating:** 7
**Confidence:** 3

**Summary:**

In this paper, authors propose to learn attribute-aware hash codes for fine-grained image retrieval, which establish the explicit correspondences between hash codes and visual attributes.

The attention mechanism is employed to extract the fine-grained tailored visual patterns from both global-level and local-level, and unsupervised reconstruction task is defined to distill the attribute related vectors through an encoder-decoder structure. Finally the hash codes are generated by the attribute vectors.

**Limitations And Societal Impact:**

No other significant concerns raise from my perspective in this round. The comments from other reviewers are very much appreciated to make further evaluation.

**Main Review:**

- The idea of distilling the high-level attribute-specific vectors in an unsupervised learning manner is interesting, and the proposed A2-Net is also equipped with the feature decorrelation constraint to enhance the representation ability of the attribute vectors.

- The experimental results on five fine-grained benchmark datasets validate the proposed method.

- The derivation of hash code generation is compact. And experimental results against state-of-the-art methods demonstrate its superiority. The additional studies are comprehensive, which includes the investigation of length of hash codes and qualitative analysis.

Post rebuttal feedback:
After considering the reviews from other reviewers and the feedback from the authors, I'm willing to raise my score to accept.

**Time Spent Reviewing:**

12

---

> ### Author Response · Authors · 2021-08-05
> **Thank you for the positive comments.**
>
> We sincerely thank you for your time and great efforts.

---

### Official Review · Reviewer_yNgD · 2021-07-16

**Rating:** 6
**Confidence:** 4

**Summary:**

This paper presents an attribute-aware hashing network for generating attribute-aware hash codes. The proposed method includes two keys components: (1) visual feature extraction with attention, (2) learning attribute-specific features with autoencoder.

The authors conducted experiments on many datasets, and it outperforms recent SOTA approaches.

**Main Review:**

Strength:
- The proposed framework (feature learning module + binary code module) is technically valid.
- Compelling results on multiple benchmark. The performance improvements, especially on CUB200, are very strong.
- Good component analysis to justify the proposed method.

Weakness:
- Unsupervised Attribute-Guided Learning: I think the proposed method is (almost) the same as learning latent codes in the general unsupervised learning literature. It is unclear to me why it is called "attribute space". It looks like it doesn't have any learning objectives that can guarantee the attribute learning. How do you ensure they are attributes? In my opinion, it is just learning latent features, which is commonly studied in the field. There is no novelty here. In other words, the term "Attribute-Guided Learning" seems overclaimed in the paper. It would be better to provide more in-depth analysis (including both quantitative and qualitative) to better justify the learned attribute features.
- Figure 1 is misleading. There is no visual question-answering components in the proposed network to help guide the unsupervised attribute learning. Again, in my opinion, it is overclaimed. The authors only use an autoencoder to obtain latent features and then use the latent features for binary code learning.
- The hash code learning module require pair-wise supervised training. One possible weakness is that, with the similarity matrix, it may not be scalable to very large dataset due to O(N^2) computational complexity.

Overall:
- The paper presents very strong results on the benchmark. However, the presentation seems overclaimed. The authors claimed that it is an unsupervised attributed-guided learning approach, but I only found latent feature learning. There is no objectives to guarantee or guide the model to understand the attributes.
- Novelty is weak. The autoencoder is commonly used to learn latent features. The proposed binary code learning module is also commonly explored in the field. Overall, I couldn't find strong novelty in the paper.

Post rebuttal:
- After discussions with other reviewers, I am a bit more convinced regarding the novelty and the contribution of the paper. Thus, I am upgrading my rating 4 --> 6.
- I am OK to accept the paper. But I still have the concern about the attribute learning. It looks like it doesn't have any learning objectives that can guarantee the attribute learning. How can we believe it is learning attribute-aware hash codes? The authors have tried to validate their claims for the attribute-aware hash codes using qualitative visualization. But the qualitative results are usually subjective. I am not sure if it is cherry-picked, but in general it is difficult to evaluate by just looking at the qualitative analysis. In other words, their claims are not thoroughly validated. It remains unclear here.

**Time Spent Reviewing:**

10

---

> ### Author Response · Authors · 2021-08-05
> **Thank you for the comments. Below please find our responses to some specific comments.**
>
> *Comment_1:  I think the proposed method is (almost) the same as learning latent codes in the general unsupervised learning literature. It is unclear to me why it is called ''attribute space''.*
>
> Response_1: The autoencoders in the general unsupervised learning literature focus on learning features, or latent codes, in an unsupervised manner. On the contrary, our approach is designed to keep the main characteristic (a.k.a. certain kinds of properties of fine-grained objects) of the encoder-decoder structure network as a projection vector, i.e., the ability to reconstruct the input signal. More importantly, the projection vectors from the visual feature representation of an image to a semantic embedding space are then employed for evaluating the fine-grained category similarities, which, as supervision, urges these projection vectors to distill the most useful properties of fine-grained objects, such as fine-grained attributes. Moreover, as validated in Figure 4 of the paper, each bit of our learned hash codes can strongly correspond to a certain kind of fine-grained object's properties, i.e., fine-grained attributes. That is the reason why such a semantic embedding space is called the attribute space in our work.
> ***
> *Comment_2: About the term "Unsupervised Attribute-Guided Learning".*
>
> Response_2: We would like to clarify that, in fact, "unsupervised attribute-guided learning" is meant to express that it is expected to learn attribute-aware vectors in an unsupervised learning manner without any explicit supervision (e.g., attribute labels or part-level annotations). We will modify this statement in the final version (perhaps "unsupervised attribute-oriented learning" could be better?). Thank you for the comment.
> ***
> *Comment_3: It would be better to provide more in-depth analysis (including both quantitative and qualitative) to better justify the learned attribute features.*
>
> Response_3: For the qualitative analyses about the learned attribute features, we have shown them in Figure 4 of the paper, which can justify the learned attribute features. Regarding the quantitative experiments, we will follow the suggestion and provide in-depth analyses in the final version. Thank you.
> ***
> *Comment_4: In Figure 1, there is no visual question-answering components in the proposed network to help guide the unsupervised attribute learning.*
>
> Response_4: In Figure 1, we originally used this as an example to illustrate what is the specific meaning of fine-grained objects' attributes, e.g., "has a red head" or "does not have a blue tail", rather than visual question-answering components. We will modify this figure for more clarification in the final version.
> ***
> *Comment_5: The hash code learning module require pair-wise supervised training. One possible weakness is that, with the similarity matrix, it may not be scalable to very large dataset due to O(N^2) computational complexity.*
>
> Response_5: As stated in Ln. 231-233 of the paper, we follow ADSH [17] by performing an *asymmetric* deep hashing training fashion, which is efficient and scalable compared with the traditional symmetric pair-wise supervised training. More specifically, in our learning procedure, we learn a deep hash function only for query points (cf. Ln. 226 of the paper), while the binary codes for database points are directly learned. Therefore, the complexity of deep hash function learning is $\mathcal{O}(M)$, where $M$ is the number of query points; and the complexity of binary codes learning is $\mathcal{O}(N)$. We will add more detailed proof in the supplementary of the final version.
> ***
> *Comment_6: About novelty.*
>
> Response_6: We would like to kindly emphasize that our main contribution is to propose $A^2$-Net as *a unified framework* to generate attribute-aware hash codes for large-scale fine-grained retrieval. In particular, the encoder-decoder network of our unsupervised reconstruction has not been explored as a fine-grained attribute distillation module before, which is novel and important in the context of large-scale fine-grained retrieval (cf. paper strengths of Reviewer iHzy). Moreover, as validated in experiments, our $A^2$-Net not only brings significant retrieval accuracy improvements but also makes the hash bits have strong correspondence to semantic visual properties (cf. Figure 4 of the paper). This kind of semantical correspondence is a new observation in related research (especially for fine-grained retrieval), which also offers an intuitive way of deep (hashing) model interpretation.

---

### Official Review · Reviewer_CEYX · 2021-07-16

**Rating:** 7
**Confidence:** 5

**Summary:**

(1)	This paper aims to address the large-scale fine-grained image retrieval, and it introduces an attention mechanism into a CNN backbone to capture fine-grained local patterns in an image. And it integrates the attentive local-level feature and the extracted global-level feature by the same CNN as the holistic feature of an image,
(2)	It proposes an Attribute-Aware Hash Codes Generating module with an encoder-decoder network by encoding the holistic feature to a latent space, and decoding the latent feature to the holistic feature, and projecting the latent feature to a binary hash code.
(3)	It designs the empirical experiments on 5 fine-grained benchmark datasets to verify the performance improvement of the proposed method compared to existing methods.


**Limitations And Societal Impact:**

No, but the authors should provide the limitations and potential negative societal impact of the work.

**Main Review:**

This paper proposes an Attribute-Aware hashing Network, namely A2-NET, to address the large-scale fine-grained image retrieval. The A2-NET integrates an attention mechanism and an encoder-decoder network with a CNN backbone, to learn attribute-aware hash codes, And the experimental results on 5 fine-grained benchmark datasets show improvement of the proposed A2-NET compared to existing methods. However, I still have some concerns as following:
(1)	The novelty of this paper is not so good because both the attention mechanism and the encoder-decoder network have been over-studied in deep learning [8, 19].
(2)	 The authors do not provide the theoretical proof to prove that the proposed A2-NET   can learn attribute-aware hash codes. And part 4.5 Qualitative Analyses of Attribute-Aware Hash Codes should provide comparisons with other methods and prove that other methods are non-attribute-aware methods.
(3)	The part Attribute-Specific Feature Decorrelation uses an orthogonal regularization, which is common in deep feature learning [a] and deep hashing [b].
(4)	There are some shortcomings about the writing, such as “In particularly” in line 27 and 112, “the explosive growing of fine-grained data” in line 27, “visual representation with attentions” in line 314, “visual attribute have been” in line 318, etc.

[a] “Can we gain more from orthogonality regularizations in training deep networks? ", NeurIPS 2018
[b] “Deep Multimodal Hashing with Orthogonal Units”. IJCAI, 2015


**Time Spent Reviewing:**

6

---

> ### Author Response · Authors · 2021-08-05
> **Thank you for the positive comments. Below please find our responses to some specific comments.**
>
> *Comment_1: Both the attention mechanism and the encoder-decoder network have been over-studied in deep learning. The part Attribute-Specific Feature Decorrelation uses an orthogonal regularization, which is common.*
>
> Response_1: We would like to kindly emphasize that our main contribution is to propose $A^2$-Net as *a unified framework* to generate attribute-aware hash codes for large-scale fine-grained retrieval. Thus, as a module of acquiring local patterns, attention mechanisms can be replaced by any other local feature extraction methods, e.g., learning convolutional filter banks as discriminative fine-grained part detectors [Ref1]. In the implementations of $A^2$-Net, we just used a vanilla attention method to verify the effectiveness of our framework. Additionally, although the encoder-decoder network has been studied in other computer vision tasks, it has not been explored as a fine-grained attribute distillation module before, which is novel and important in the context of large-scale fine-grained retrieval (cf. paper strengths of Reviewer iHzy). Moreover, as validated in experiments, such an unsupervised reconstruction design of $A^2$-Net brings significant retrieval accuracy improvements, as well as making the hash bits have strong correspondence to semantic visual properties. This kind of semantical correspondence is a new observation in related research (especially for fine-grained retrieval, cf. Figure 4 of the paper), which also offers an intuitive way of deep hashing interpretation.
>
> [Ref1] Y. Wang, V. I. Morariu and L. S. Davis. Learning a Discriminative Filter Bank within a CNN for Fine-grained Recognition. CVPR 2018.
> ***
> *Comment_2: Part 4.5 Qualitative Analyses of Attribute-Aware Hash Codes should provide comparisons with other methods and prove that other methods are non-attribute-aware methods.*
>
> Response_2: Thank you for the comment. During experiments, we indeed conducted the same qualitative analyses for comparisons with other methods and found other methods are non-attribute-aware methods. However, we cannot show them in the OpenReview system in the rebuttal phase. We will add these comparisons in the final version by following your constructive suggestion.
> ***
> *Comment_3: There are some shortcomings about the writing.*
>
> Response_3: Thank you for pointing out these issues. We will carefully proofread the paper in the final version.

---

### Official Review · Reviewer_iHzy · 2021-07-16

**Rating:** 9
**Confidence:** 5

**Summary:**

This paper proposes a unified framework to generate attribute-aware hash codes for dealing with the large-scale fine-grained image retrieval task. Specifically, it develops an attribute-aware hashing network, termed as A^2-Net, consisting of two main modules including fine-grained representation learning and attribute-aware hash codes generating. In A^2-Net, the proposed reconstruction task realizing by an encoder-decoder structure is proven to be able to unsupervisedly distill high-level attribute-specific vectors from the appearance visual representations without attribute annotations. Thus, the hash codes can be generated by the attribute vectors based on a feature decorrelation constraint and the similarity preserving constraint. Experiments are conducted on several fine-grained benchmarks, which can show outperforming on both qualitative and quantitative aspects.

**Limitations And Societal Impact:**

The attribute-aware hash code in this paper is interesting. The authors also have shown some visualization results. But, I am still wondering that what is the relation between the attention in the feature learning module and the generated hash codes? Can they correspond one by one? The authors are encouraged to discover that in the future.

**Main Review:**

+ The large-scale fine-grained retrieval problem studied in this paper is a fundamental and practical task that deserves further study.
+ This paper not only develops an effective method wrt retrieval accuracy, but also brings interesting interpretability. The whole method is well-motivated.
+ The technique contributions of this paper are novel and reasonable, which is tailored for the challenges of large-scale fine-grained retrieval. Particularly, the proposed encoder-decoder structure for distilling high-level attribute-specific vectors from visual features is interesting. Moreover, the feature decorrelation constraint is introduced for further improving the discriminative ability of these attribute vectors.
+ Experiments are conducted on five popular fine-grained datasets, which show consistent and significant improvements over baseline methods. Also, the ablation studies are adequate and can validate the effectiveness and robustness of A^2-Net, as well as its main components.
+ The qualitative analyses of these attribute-aware hash codes in Sec. 4.5 validate the effectiveness of the proposed method and offer good interpretation.
+ This paper is well written and easy to follow.


**Time Spent Reviewing:**

3

---

> ### Author Response · Authors · 2021-08-04
> **Thank you for the positive comments. Below please find our responses to some specific comments.**
>
> *Comment_1: I am still wondering that what is the relation between the attention in the feature learning module and the generated hash codes? Can they correspond one by one? The authors are encouraged to discover that in the future.*
>
> Response_1: As stated in Ln. 272 of the paper, in our $A^2$-Net, the number of attention guidance is set to equal the number of hash bits. Thus, they are designed to correspond to each other. In the future, we will delve deep into the neural weights between the attentions and the generated hash codes to clarify their relationship from both quantitative and qualitative aspects. Thank you for the constructive comment.

---

### Decision · Program_Chairs · 2021-09-27

**Decision:**

Accept (Spotlight)

**Comment:**

Thanks for your submission to NeurIPS.

The reviewers were all in agreement that this is a solid paper that deserves to be accepted.  Overall, the reviewers praised the importance of the problem, the proposed solution, and the empirical results.  After the rebuttal and discussion, all advocated for accepting the paper.  Despite being overall very positive about the paper, the reviewers did note a few weaknesses; please keep these in mind when preparing a final version of the manuscript.